# Molecular evidence of hybridization between pig and human *Ascaris* indicates an interbred species complex infecting humans

Alice Easton[1,2†], Shenghan Gao[3,4†‡], Scott P Lawton[5], Sasisekhar Bennuru[1], Asis Khan[6], Eric Dahlstrom[7], Rita G Oliveira[2], Stella Kepha[8], Stephen F Porcella[7], Joanne Webster[2,9], Roy Anderson[2], Michael E Grigg[6], Richard E Davis[3]*, Jianbin Wang[3,10]*, Thomas B Nutman[1]*

[1]Helminth Immunology Section, Laboratory of Parasitic Diseases, National Institute of Allergy and Infectious Disease, National Institutes of Health, Bethesda, United States; [2]Department of Infectious Disease Epidemiology, Imperial College London, London, United Kingdom; [3]Department of Biochemistry and Molecular Genetics, RNA Bioscience Initiative, University of Colorado School of Medicine, Aurora, United States; [4]Beijing Institute of Genomics, Chinese Academy of Sciences, Beijing, China; [5]Epidemiology Research Unit (ERU) Department of Veterinary and Animal Sciences, Northern Faculty, Scotland's Rural College (SRUC), Inverness, United Kingdom; [6]Molecular Parasitology Section, Laboratory of Parasitic Diseases, National Institute of Allergy and Infectious Disease, National Institutes of Health, Bethesda, United States; [7]Genomics Unit, Research Technologies Branch, National Institute of Allergy and Infectious Diseases, National Institutes of Health, Hamilton, United States; [8]London School of Tropical Medicine and Hygiene, London, United Kingdom; [9]Royal Veterinary College, University of London, Department of Pathobiology and Population Sciences, Hertfordshire, United Kingdom; [10]Department of Biochemistry and Cellular and Molecular Biology, University of Tennessee, Knoxville, United States

*For correspondence:
RICHARD.DAVIS@cuanschutz.edu
(RED);
jianbin.wang@utk.edu (JW);
tnutman@niaid.nih.gov (TBN)

†These authors contributed
equally to this work

Present address: ‡Institute of
Microbiology, Chinese Academy
of Sciences, Beijing, China

Competing interest: See
page 18

Reviewing editor: Nicola L
Harris, Monash University,
Australia

**Abstract** Human ascariasis is a major neglected tropical disease caused by the nematode *Ascaris lumbricoides*. We report a 296 megabase (Mb) reference-quality genome comprised of 17,902 protein-coding genes derived from a single, representative *Ascaris* worm. An additional 68 worms were collected from 60 human hosts in Kenyan villages where pig husbandry is rare. Notably, the majority of these worms (63/68) possessed mitochondrial genomes that clustered closer to the pig parasite *Ascaris suum* than to *A. lumbricoides*. Comparative phylogenomic analyses identified over 11 million nuclear-encoded SNPs but just two distinct genetic types that had recombined across the genomes analyzed. The nuclear genomes had extensive heterozygosity, and all samples existed as genetic mosaics with either *A. suum*-like or *A. lumbricoides*-like inheritance patterns supporting a highly interbred *Ascaris* species genetic complex. As no barriers appear to exist for anthroponotic transmission of these 'hybrid' worms, a one-health approach to control the spread of human ascariasis will be necessary.

## Introduction

Approximately 447 million people were estimated to be infected with the intestinal nematode *Ascaris lumbricoides* in 2017, resulting in an estimated 3206 deaths and a loss of over 860,000 Disability-Adjusted Life Years (DALYs, *Global Burden of Disease Study, 2017*; http://ghdx.healthdata.org/gbd-2017). Many infections go undiagnosed, but like other soil-transmitted helminths (STH), *Ascaris* spp. infections contribute significantly to global DALYs, perpetuating the cycle of poverty in areas of endemic infection (*Brooker, 2010*; *Hotez et al., 2008*; *Montresor et al., 2012*; *Pullan et al., 2014*). Despite the large global burden of STH, little is known about *A. lumbricoides* transmission patterns or the true prevalence of infection with the pig parasite *A. suum* infection in humans in endemic regions.

Deworming has become more widespread in areas of endemic STH infection (*Bundy et al., 2017*). Regional health authorities and global health organizations are now looking for strategies to build on these programs by achieving local elimination of STH as a public health problem (*Becker et al., 2018*). A greater understanding of transmission dynamics (including the frequency of zoonotic transmission) using molecular epidemiological methods in settings where *A. lumbricoides* prevalence is low but persistent could help move current efforts toward successfully eliminating transmission through more targeted treatment.

Population genetic studies of *A. lumbricoides* have drawn varying conclusions about whether zoonotic transmission is frequent (*Anderson and Jaenike, 1997*; *Dutto and Petrosillo, 2013*; *Nejsum et al., 2012*; *Nejsum et al., 2005a*). Some studies have shown that cross-species transmission occurs between pigs and humans living in close proximity (*Anderson, 1995*; *Betson et al., 2014*; *Miller et al., 2015*; *Monteiro et al., 2019*; *Nejsum et al., 2005b*; *Peng and Criscione, 2012*; *Sadaow et al., 2018*; *Takata, 1951*; *Zhu et al., 1999*). This is especially common in non-endemic regions, probably because zoonotic transmission is less likely to be identified in areas where human-to-human transmission is common. The human parasite *A. lumbricoides* and the pig parasite *A. suum* have been found to be capable of interbreeding, and 4–7% of worms in Guatemala and China were hybrids (*Criscione et al., 2007*; *Peng and Criscione, 2012*). Furthermore, it is unclear whether pigs are an important reservoir of infection in humans worldwide or if *A. suum* is readily transmitted anthroponotically (*Betson et al., 2013*; *Betson and Stothard, 2016*; *da Silva Alves et al., 2016*; *Leles et al., 2012*; *Nejsum et al., 2012*). Studies have generally concluded that the genetic differences between *Ascaris* worms collected from human populations in different parts of the world (*Betson et al., 2014*; *Peng et al., 1998*) are the result of geographic reproductive isolation. Previous studies using *Ascaris* mitochondrial genomes or genes suggest there are *A. lumbricoides*-type (human-associated) and *A. suum*-type (pig-associated) clades (*Anderson and Jaenike, 1997*; *Cavallero et al., 2013*; *Zhou et al., 2011*). Other work suggests multiple clades of worms, only one of which is unique to pigs (*Nejsum et al., 2017*). *Ascaris* spp. infections also occur naturally in monkeys and apes, and *Ascaris* spp. eggs are sometimes found in the feces of dogs but this is likely a result of coprophagy by the dogs, rather than due to infection (https://www.cdc.gov/parasites/ascariasis/biology.html).

In the current study, we constructed a reference-quality *Ascaris* genome (ALV5) based on sequences from a single female worm collected from a person in Kenya. This person was presumed to be infected with *A. lumbricoides* as there is a lack of local pig husbandry. Draft *A. suum* genomes have previously been constructed from worms obtained from pigs in Australia (*Jex et al., 2011*) and in the United States (*Wang et al., 2011*; *Wang et al., 2012*). The *Ascaris* genome ALV5 was found to be highly similar (99% identity) to the *A. suum* genome from worms collected from pigs in the United States (*Wang et al., 2017*). Our mitochondrial and whole-genome analyses from an additional 68 individual worms indicate that *A. suum* and *A. lumbricoides* form a genetic complex that is capable of interbreeding. Our data support a model for a recent worldwide, multi-species *Ascaris* population expansion caused by the movement of humans and/or livestock globally. *Ascaris* from both pigs and humans may be important in human disease, necessitating a one-health approach to control the spread of human ascariasis.

## Results

### Human *Ascaris* reference genome to promote comparative genomic analyses

To generate a human *Ascaris* spp. germline genome assembly (prior to programmed DNA elimination *Wang et al., 2017*), ovarian DNA was sequenced from a single female worm collected from a Kenyan study participant who was presumed to be infected with *A. lumbricoides* using Illumina paired-end and mate-pair libraries of various insert sizes with a total sequence coverage of ~27 fold (*Supplementary file 1*). Using these data, three different assembly strategies were used. The de novo assembly and semi-de novo strategies produced poor *A. lumbricoides* germline draft genomes (*Table 1*). In the semi-de novo assembly, the majority of the >4000 short contigs (making up 15.4 Mb of sequence) that could not be incorporated into the semi-de novo assembly are sequences that aligned to the genome at multiple positions. Comparison of the *A. suum* gene annotations to this assembly revealed a low *A. lumbricoides* gene number and high numbers of partial and split genes (*Table 1*, see footnote 3). These characteristics are typical of highly fragmented genomes or genomes with high levels of mis-assemblies (*Wang et al., 2017*).

Mapping of the human *Ascaris* reads to the *A. suum* reference genome (*Wang et al., 2017*) revealed an exceptionally high-sequence similarity (>99% identity) between the two species with few human *Ascaris* reads that could not be mapped to *A. suum*. Based on this high-sequence similarity, a third reference-based-only assembly strategy was used to generate the human *Ascaris* germline genome assembly using the *A. suum* germline genome as a reference (see Materials and methods). This approach led to a reference-quality human *Ascaris* genome assembly with many fewer gaps (only 0.98 Mb of sequence) and no unplaced contigs. The *Ascaris* genome assembled into 415 scaffolds with a combined size of 296 Mb. An additional 15.4 Mb of sequence was present in 4072 unscaffolded short contigs. The assembly N50 value was 4.63 Mb, with the largest scaffold measuring 13.2 Mb. The largest 50 scaffolds combined to represent 78% of the genome. The assembly was further polished using additional Illumina reads from the same worm to more accurately reflect single base differences, indels, and any potential local mis-assembled regions.

To evaluate the quality of the assembled genome, we mapped the *Ascaris* Illumina reads back to the reference-based *Ascaris* genome assembly and found that >99% of the Illumina reads could be mapped, indicating that the reference-based assembly excluded very few *Ascaris* reads. We then mapped and transferred the extensive set of *A. suum* transcripts (*Jex et al., 2011*; *Wang et al., 2017*) to the human *Ascaris* germline assembly to annotate the genome, identifying and classifying 17,902 protein-coding genes (*Table 1*, *Supplementary file 1*). As this reference-based assembly

**Table 1.** *Ascaris* germline genome assemblies.

| Features | *A. lumbricoides* de novo | *A. lumbricoides* semi-de novo[*] | *A. lumbricoides* reference-based | *A. suum*[†] (*Wang et al., 2017*) | *A. suum*[†] (*Jex et al., 2011*)[§] |
|---|---|---|---|---|---|
| Assembled bases (Mb) | 269.2 | 307.9 | 296.0 | 298.0 | 272.8 |
| N50 (Mb) | 0.29 | 4.77 | 4.63 | 4.65 | 0.41 |
| N50 number | 269 | 21 | 21 | 21 | 179 |
| N90 (Mb) | 0.04 | 0.95 | 0.91 | 0.92 | 0.08 |
| N90 number | 1112 | 74 | 75 | 75 | 748 |
| Total scaffold number | 8111 | 412 | 415 | 415 | 29,831 |
| Largest scaffold length (Mb) | 1.9 | 13.9 | 13.2 | 13.4 | 3.8 |
| Protein-coding genes | 17,011[‡] | 17,105[‡] | 17902 | 18,025 | 18,542[‡] |

[*] Exhibits ~23 Mb of sequence gaps and 15.4 Mb of unplaced sequence in 4072 short contigs.

[†] The three *A. lumbricoides* assemblies constructed here are compared to the *A. suum* assemblies from Australia (*Jex et al., 2011*) and the United States (*Wang et al., 2017*).

[‡] 21–23% are only partial genes based on the annotation from *A. suum* (*Wang et al., 2017*).

[§] The sample for sequencing is derived from a mixture of the germline and somatic genomes (after DNA elimination).

exhibits the best assembly attributes, including high continuity with a large N50, low gaps and unplaced sequences, and high-quality protein-coding genes (see *Table 1*), we suggest that this version should be used as a reference germline genome for a human *Ascaris* spp. specimen (available in NCBI GenBank with accession number PRJNA515325). The other two assemblies are available online.

Like *A. suum* embryos, *A. lumbricoides* embryos undergo programmed DNA elimination during the differentiation of the somatic cells from the germline in early development (*Streit et al., 2016*; *Wang and Davis, 2014*). In *A. suum*, ~30 Mb of 120 bp tandem repeats and ~1000 germline-expressed genes are lost from the germline to form the somatic genome (*Wang et al., 2012*; *Wang et al., 2017*). We also sequenced the somatic genome from the intestine of the same female *A. lumbricoides* worm. Comparison of the germline and somatic genomes revealed that DNA elimination in the human *Ascaris* sample (including the breaks, sequences, and genes eliminated) was identical to that described for the pig *A. suum* sample (*Wang et al., 2017*).

## Gene content and *Ascaris* proteome

Earlier annotations of protein coding genes for *A. suum* draft genomes were produced by *Jex et al., 2011* and *Wang et al., 2012* and improved with a recent updated genome (*Wang et al., 2017*)—although the focus of the recent study was not on protein annotations. Here, we updated, identified, and fully annotated the 17,902 protein-coding genes in the reference-based genome assembly (*Supplementary file 2* and *Figure 1—figure supplement 1*). Our aims were to highlight the phylogenetic relationship with other helminths and between *Ascaris* spp., to provide potential targets for future diagnostics to differentiate between nematodes and even between pig and human *Ascaris*, and to detail the potential functions of hypothetical or unknown proteins in the *Ascaris* genome. Using a custom pipeline (see Materials and methods and *Cotton et al., 2017*), we classified 48% of the predicted proteome into functional groups (*Figure 1A*). Although the remaining 52% (9300) of the genes were classified as unknown/uncharacterized, 2515 (27%) of these appear to encode proteins that have signatures indicative of either being secreted or being membrane-bound (some with GPI anchors). To provide a more comprehensive annotation of the transcriptomes of *A. suum* and *A. lumbricoides*, we re-mapped the RNA-seq data from *A. suum* to the current gene models of *A. lumbricoides* (ALV5) (*Supplementary file 2*). We performed multivariate analyses of this revised RNA-seq data compilation to generate a comprehensive RNA-seq data set for differential gene expression in diverse stages/tissues (*Supplementary file 2*).

Phylogenetic trees derived from orthologue analyses of the predicted proteomes of ALV5 with the predicted proteomes of other nematodes across all clades indicated the similarity among the published genomes of *A. suum* PRJNA62057 and PRJNA80881 in *Jex et al., 2011*; *Wang et al., 2012*; *Wang et al., 2017* and *A. lumbricoides* (*International Helminth Genomes Consortium, 2019*) with ALV5 within the *Ascaris* branch (*Figure 1C*). The variation observed within the *Ascaris* spp. (with relatively weak bootstrap values of 0.3–0.59) is likely due to the differences in protein coding gene annotations and split genes seen in previous assemblies.

## Mitochondrial genome assembly

We next took advantage of the abundant reads from the mitochondrial genome in our sequencing data (on average 7690X coverage, see *Supplementary file 1*) to perform de novo assembly of 68 complete human *Ascaris* spp. mitochondrial genomes from individual worms (*Supplementary file 3*). These mitochondrial genomes were then annotated using sequence similarity to well-characterized and annotated mitochondrial genes.

## Population structure inferred from mitochondrial cox-1 gene

The mitochondrial cox-1 gene has been frequently used to infer evolutionary distances between species as well as between populations (*Cavallero et al., 2013*; *Amor et al., 2016*; *Springer et al., 2001*; *Wiens et al., 2010*; *Zardoya and Meyer, 1996*; *Zou et al., 2017*) due to its rapid mutation rate, lack of recombination and relatively constant rate of change over time (*Brown et al., 1979*; *Giles et al., 1980*; *Harrison, 1989*). Existing data suggest that mitochondria are inherited maternally in *C. elegans* (*Lim et al., 2019*; *Zhou et al., 2011*; *Sato and Sato, 2011*; *Wang et al., 2017*) and Ascaris (*Anderson et al., 1995*). Previous cox-1 phylogeny studies resolve *Ascaris* spp. worms into

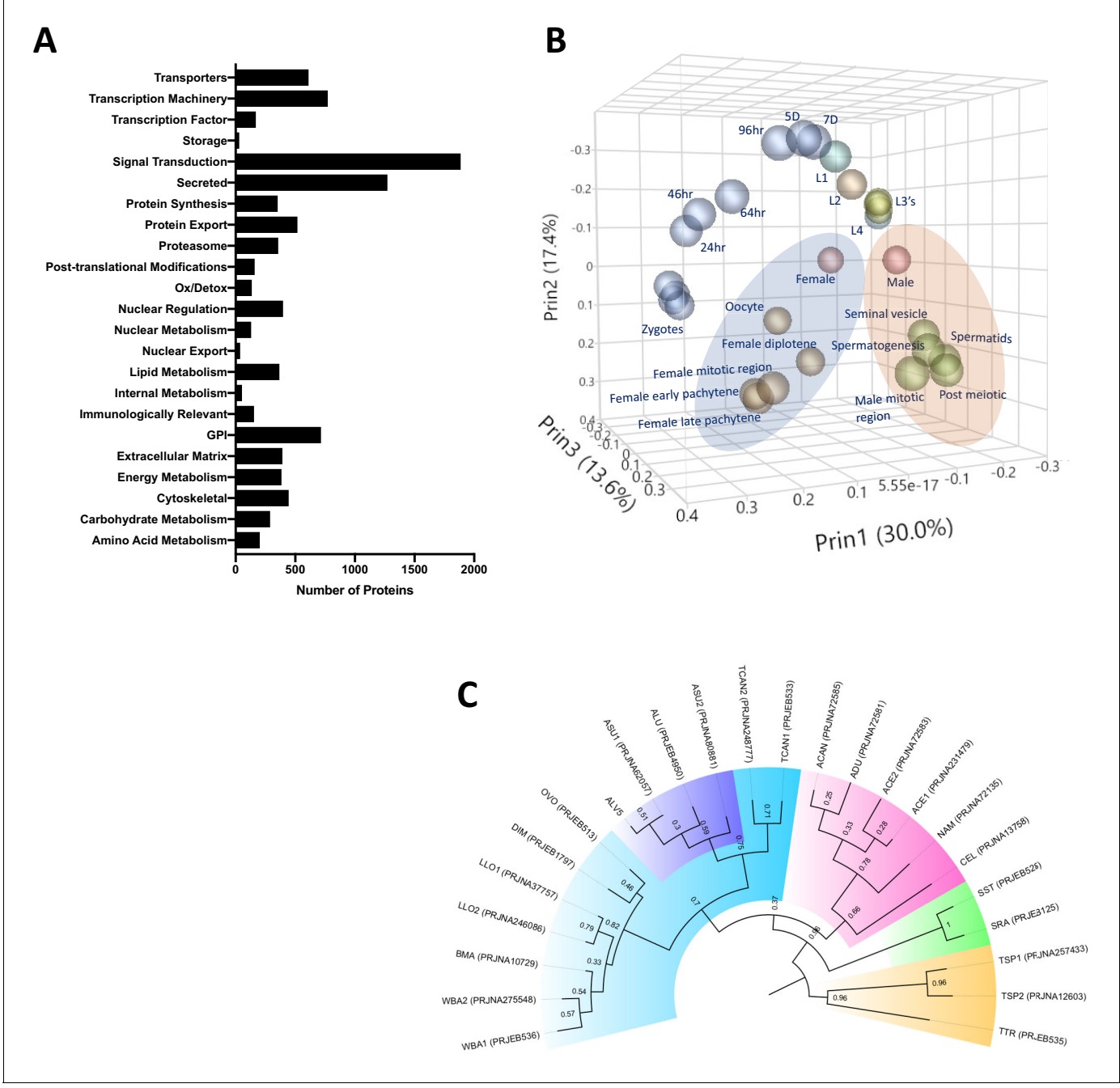

**Figure 1.** *Ascaris* proteome. (**A**) Functional classification of the predicted proteome of *A. lumbricoides* (an improved proteome of *Ascaris* spp.), excluding proteins with unknown or uncharacterized function. (**B**) PCA plot based on multivariate analyses of RNA-seq data from various stages/tissues. Samples from tissues related to sperm (blue ellipse) and oocyte production (orange ellipse, see also *Figure 1—figure supplement 2*) cluster together. (**C**) Estimated tree based on orthology analyses between the predicted proteomes of publicly available nematodes. The *Ascaris* clade has been shaded in purple within Clade III (teal). Samples are labeled by BioProject Accession number, as well as by the first letter of the genus and the first two letters of the species name (ASU = *Ascaris suum*, ALU = *Ascaris lumbricoides*, WBA = *Wuchereria bancrofti*, BMA = *Brugia malayi*, LLO = *Loa loa*, DIM = *Dirofilaria immitis*, OVO = *Onchocerca volvulus*, TCAN = *Toxocara canis*, ACAN = *Ancylostoma caninum*, ADU = *Ancylostoma duodenale*, ACE = *Ancylostoma ceylanicum*, NAM = *Necator americanus*, CEL = *Caenorhabditis elegans*, SST = *Strongyloides stercoralis*, SRA = *Strongyloides ratti*, TSP = *Trichinella spiralis*, TTR = *Trichuris trichiura*). Multiple genomes for the same organism are suffixed with numerals.
The online version of this article includes the following figure supplement(s) for figure 1:

**Figure supplement 1.** Predicted proteome and stage-specific transcriptomes of Ascaris.

**Figure supplement 2.** *Ascaris* stage-specific RNA expression heatmaps.

three distinct clades: clade A is predominantly comprised of worms isolated from pigs, clade B is predominantly comprised of worms isolated from humans, and clade C is from worms only isolated from pigs in Europe and Asia (*Cavallero et al., 2013*). Interestingly, haplotype network analyses revealed that the majority of worms isolated from humans in the Kenyan villages possessed cox-1 haplotypes that were consistent with infection of parasites from clade A (63/68), whereas only six specimens had cox-1 haplotypes consistent with infection by worms from clade B (*Figure 2—figure supplement 1* and *Figure 2a*).

When cox-1 sequences from the present study were compared against those within the *Ascaris* species complex deposited at NCBI (see *Supplementary file 4* and *Figure 2B*; *Cotton et al., 2017*; *Criscione et al., 2007*; *Godel et al., 2012*; *Goldberg et al., 2013*) within clade A (which appeared to contain the majority of sequences not only from Kenya but also from other localities), seven unique haplotypes of cox-1 from Kenya were identified. These appeared to be shared not only with other haplotypes from Africa, but also with those from Brazil. In contrast, clade B haplotypes appeared to be even more cosmopolitan, with the three haplotypes from Kenya not only being shared with Zanzibar, but also with haplotypes from Brazil, Denmark, China and Japan. Despite the distinct clustering of haplotypes into the three typical *Ascaris* clades, there was very little genetic diversity among haplotypes within each of the clades, with the majority of haplotypes being separated by 1–4 nucleotide differences. There were greater levels of genetic divergence between clades; A and B were closer to each other while C was more distinct. Similar findings were seen with nad-4, the most variable gene in the mitochondrial genome (*Figure 2—figure supplement 1*, *Figure 2—figure supplement 2*).

## Phylogenetic analyses and population structure inferred from complete mitochondrial genomes

Forty-seven SNPs were identified in the human *Ascaris* mitochondrial genomes. Approximately a quarter of these variants were in non-coding portions of the mitochondrial genome and half were synonymous (*Supplementary file 1*). As with the cox-1 haplotype analyses, whole mitochondrial genome analysis distinguished two clades (clade A and clade B), but there were no distinct geographically specific sub-clades seen within either clade A or clade B (*Figure 2B*, *Table 2*). Clade C was also produced by a single published sequence which was used for comparison. In order to assess the validity of the clades A and B representing two distinct molecular taxonomic units, and thus potentially different species, *Birky, 2013* 4X ratio was applied to provide a lineage-specific perspective of potential species delimitation. The ratio failed to differentiate clades A and B as distinct species with $K/\Theta < 4$ at 2.285 indicating *Ascaris* is one large population—further supporting the lack of differentiation into separate species (*Supplementary file 5*). Furthermore, there were no significant associations between mitochondrial sequence variations and other factors (e.g. village, household, time of worm collection, host) based on PERMANOVA (see methods and *Table 2*) after translating the phylogenetic tree into a distance matrix, suggesting not only a lack of differentiation into distinct species but also a potentially large interbreeding population of worms being transmitted between individuals and across villages.

To account for a potentially large population of interbreeding worms, analyses to detect signatures of population expansion were performed. When the global mitochondrial genome data were compared, the Tajima's D was negative and significant (Tajima's D −1.5691; p-value 0.028), indicating an excess of low frequency polymorphisms within the global data set suggesting population size expansion. Despite the Fu's F not being significant it was positive (Fu's Fs 8.5673; P-0.975) potentially indicating a deficiency in diversity as would be expected in populations that have recently undergone a bottleneck event. The same pattern was also seen in the Kenyan sequences but neither the Tajima's D nor the Fu's were significant (*Figure 2—figure supplement 3* and *Supplementary file 6*). Although there does appear to be a signature of a recent population expansion event in both the global and Kenyan data, the lack of information on the mutation rates of *Ascaris* and other nematodes prevents the accurate estimate of such an event.

## Nuclear genome variation in the *Ascaris* population

To quantify genetic variation in the *Ascaris* worms isolated from infected Kenyans, the nuclear genomes of the 68 individual worms were analyzed to assess intraspecific population genetic

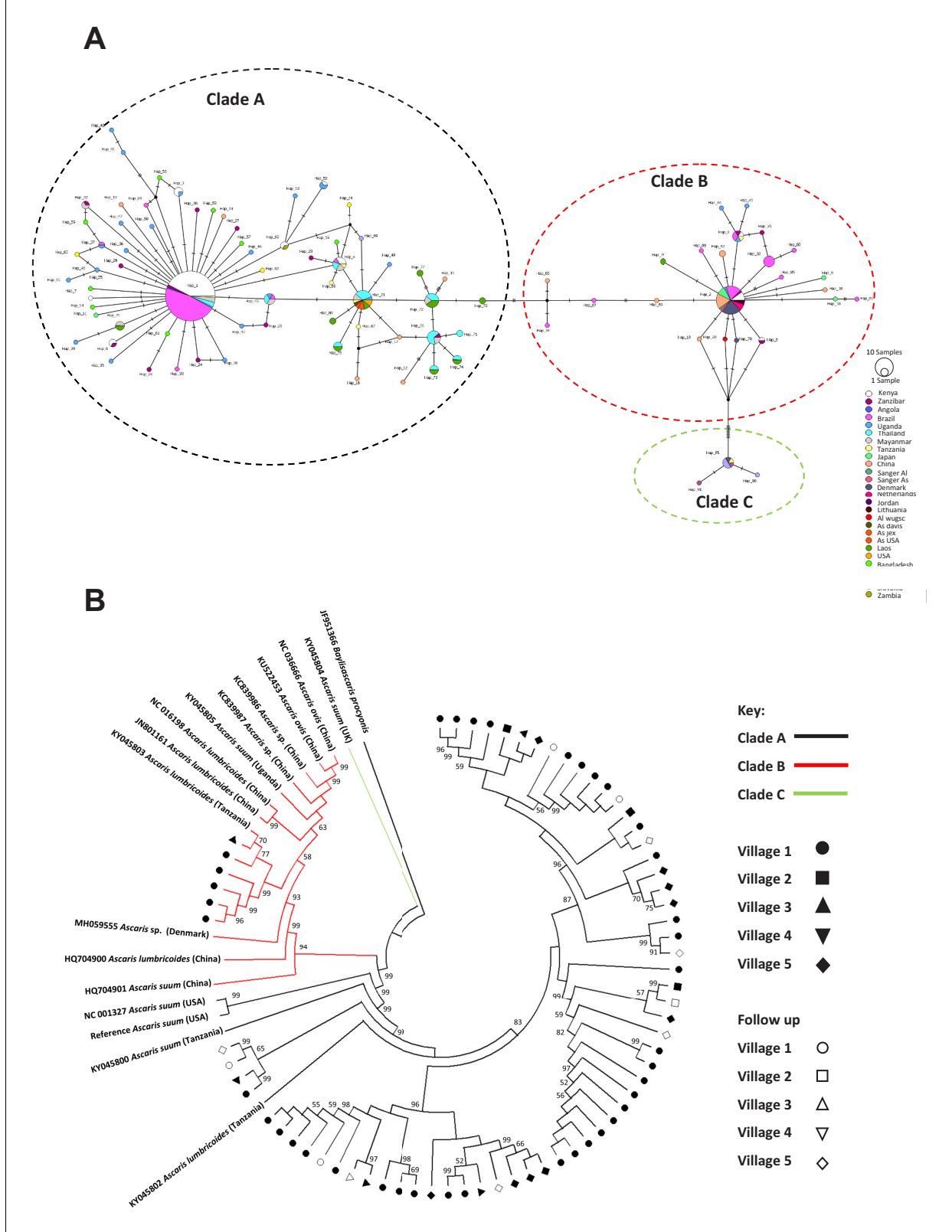

**Figure 2.** Phylogenetics of *Ascaris* spp based on mitochondrial sequences. (**A**) Haplotype network based on the COl mitochondrial gene. Notches on the lines separating samples represent the number of nucleotide changes between the worms represented, details on the origins of haplotypes can be found in *Supplementary file 4*; (**B**) Maximum likelihood phylogenetic (ML) reconstruction of *Ascaris* complete mitochondrial genomes, constructed under the conditions of the GTR model and 1000 bootstrap replicates were used to provide nodal supports. The tree was constructed using all

*Figure 2 continued on next page*

mitochondrial genomes assembled from the Kenyan worm specimens and all other published reference *Ascaris* mitochondrial genomes and *Baylisascaris procyonis* was used as the outgroup. The three major clades A, B, and C were identified by color hue, and the majority of the Kenyan worms clustered in clade A. Each village was represented by a distinct shape and unfilled shapes represented worms sequenced from specific villages post-anthelminthic treatment.

The online version of this article includes the following figure supplement(s) for figure 2:

**Figure supplement 1.** Phylogenetic trees based on cox-1 and nad-4.

**Figure supplement 2.** Sliding window analyses.

**Figure supplement 3.** Evidence of *Ascaris* population expansion.

**Figure supplement 4.** *Ascaris* SNPs and insertion/deletions (indels) maps of representative chromosomal fragments.

diversity, heterozygosity, and ploidy. Single-nucleotide polymorphisms (SNPs) and insertion/deletions (indels) across the nuclear genomes were assessed for the first 50 largest scaffolds, which comprised 78% of the genome (see methods). Each *Ascaris* worm was sequenced to a mean coverage depth of ~27 fold. A total of 11.15 million SNP positions were identified in the first 50 scaffolds among the *Ascaris* nuclear genomes. Approximately 25% of these variants were intergenic (*Supplementary file 1*). As an example, SNPs and indels in a single *Ascaris* chromosome were plotted for two worms collected from humans in Kenya and one worm from a pig in the United States (*Figure 2—figure supplement 4*). The profiles and the frequency between SNPs and indels are highly consistent within individual worms, with the ratio of indel:SNPs frequency at ~1:7. A comparison of the variations identified between individuals infected with worms that had either *A. lumbricoides*-like or *A. suum*-like mitochondrial genomes illustrates that most of the differences appear to be random variations, and there do not appear to be major differences between *A. lumbricoides*-like and *A. suum*-like worms. A total of 1.79 million SNPs were unique to individual specimens, presumably representing genetic drift. Of the remaining 9.3 million SNPs, ~32% of these variant positions were present in less than five specimens indicating that the *Ascaris* genomes sequenced are ~1% polymorphic among the major alleles circulating within the species complex.

## Population structure inferred from nuclear genomes

To investigate the evolutionary pressures that account for the high SNP diversity found among the 68 sympatric worms, the ploidy, degree of heterozygosity (*He*) and allelic diversity were determined. Worms were disomic, with little to no evidence of aneuploidy (*Figure 3—figure supplement 1*). The vast majority (>98%) of SNP positions were biallelic, and each worm had, on average, 2.3 million variant positions, of which approximately 60% were heterozygous SNPs (*Supplementary file 7*). SNP density was determined in 10 kb windows for each worm against the reference ALV5 and a patchy, mosaic pattern was resolved. SNP density was structured within the genome, with scaffolds being either SNP poor or SNP dense. For example, Algv5r020 was SNP dense whereas Algv5r019x was SNP poor. In other scaffolds, alternating SNP poor and SNP dense regions were defined within the contig, with distinct transition points, see for example the first half of Algv5b02, the last quarter of

**Table 2.** Effects of host, household, village, and time point on the genetic variation of *Ascaris*.

| | Nuclear genome phylogeny[*] | | | Mitochondrial genome phylogeny | | |
|---|---|---|---|---|---|---|
| | R[†] | p-value | p-adjusted (Bonferroni) | R[†] | p-value | Samples[†] |
| Individual | 0.933 | *0.001* | *0.004* | 0.996 | 0.095 | 68 worms from 60 people |
| Household | 0.020 | 0.110 | 0.440 | 0.011 | 0.340 | 68 worms from 43 houses |
| Village | 0.052 | *0.001* | *0.004* | 0.013 | 0.335 | Five villages with 43, 17, 4, 3, and one individual each |
| Time point | 0.018 | 0.162 | 0.648 | 0.024 | 0.100 | 55 at baseline and 13 post-deworming |

[*] Results based on PERMANOVA using phylogenetic distances among worms. Results were largely similar using a distance matrix generated from the PCA plot (*Figure 6*) and using the Multi-Response Permutation Procedure (MRPP) method (*Supplementary file 9*).

[†] Since some worms did not have metadata associated with each variable examined, and some variables were over-represented in the sample (for example, 43 of 68 worms came from a single village) the samples are specified in this column.

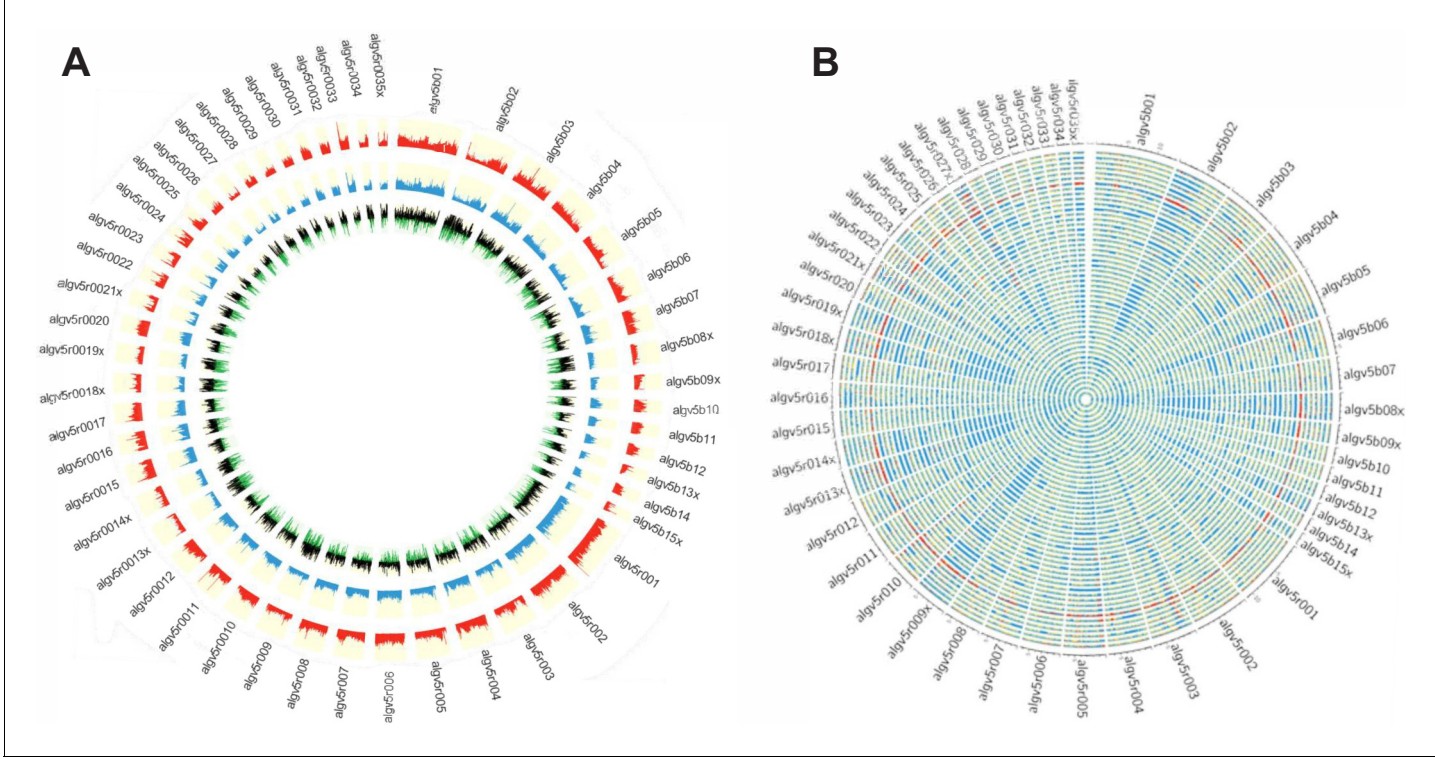

**Figure 3.** Genetic diversity of the *Ascaris* specimens. (A) Circos plot depicting the genetic diversity of the Ascaris specimens. Outside track (red histograms) shows the total SNP diversity across the genome (first 50 largest scaffolds) in 10 kb sliding windows. Blue bar plot indicates the measured degree of polymorphism (π) (*Nei and Li, 1979*) within the Ascaris population in 10 kb sliding windows. The innermost track with black-green histogram plots the *Tajima, 1989* values which reflect the difference between the mean number of pairwise differences (π) and the number of segregating sites using a sliding window of 10 kb. (B) The Circos-plot of the genome-wide distribution of heterozygous and homozygous SNPs in 10 kb blocks identified long stretches of homozygosity among the different *Ascaris* specimens, except 119_3, which is predominantly heterozygous throughout and was isolated from village 3. Red color = >90% of heterozygous SNPs, blue = >90% of homozygous SNPs, yellow = 50% heterozygous, 50% homozygous SNPs. Each track represents a single specimen.

The online version of this article includes the following figure supplement(s) for figure 3:

**Figure supplement 1.** Somy analysis of the *Ascaris* worm specimens.

Algv5b05, or the middle of Algv5r021x (*Figure 3A*). In those regions where SNP density was low, the Tajima D statistic was net negative, indicating that allele frequencies within these regions were structured and more limited.

Genome-wide, homozygous SNP regions were found to be unevenly distributed, with some scaffolds possessing long runs of homozygosity, see for example Algv5b02, Algv5r009x, Algv5r013x, Algv5r014x, Algv5r018x, Algv5r019x, Algv5r027x (depicted by solid blue in *Figure 3B*), and these regions were net negative by the Tajima D test. Conversely, heterozygous SNPs were less structured and appeared randomly distributed throughout the genome (*Figure 3B*). Overall, three genetic types were resolved by this analysis: in each genome, there existed SNP-poor homozygous regions (colored blue) or SNP dense regions, which either possessed homozygous alternate SNPs (also colored blue) or heterozygous SNPs (colored in 'red' or 'yellow' blocks depending on the density of heterozygous SNPs resolved in each 10 kb block: one haplotype was similar to ALV5 and the other was different). Only one worm specimen (119_3) was heterozygous genome-wide, and this track is depicted as 'red' across all scaffolds in the Circos plot (*Figure 3B*).

## Population genetic structure of Kenyan *Ascaris* worm specimens

A phylogenetic tree constructed using genome wide SNPs with at least 10x coverage (11.15 million phased SNPs total) from 69 Ascaris worm specimens, including the *A. suum* reference genome, established that the Kenyan specimens were more similar to each other than they were to the *A.*

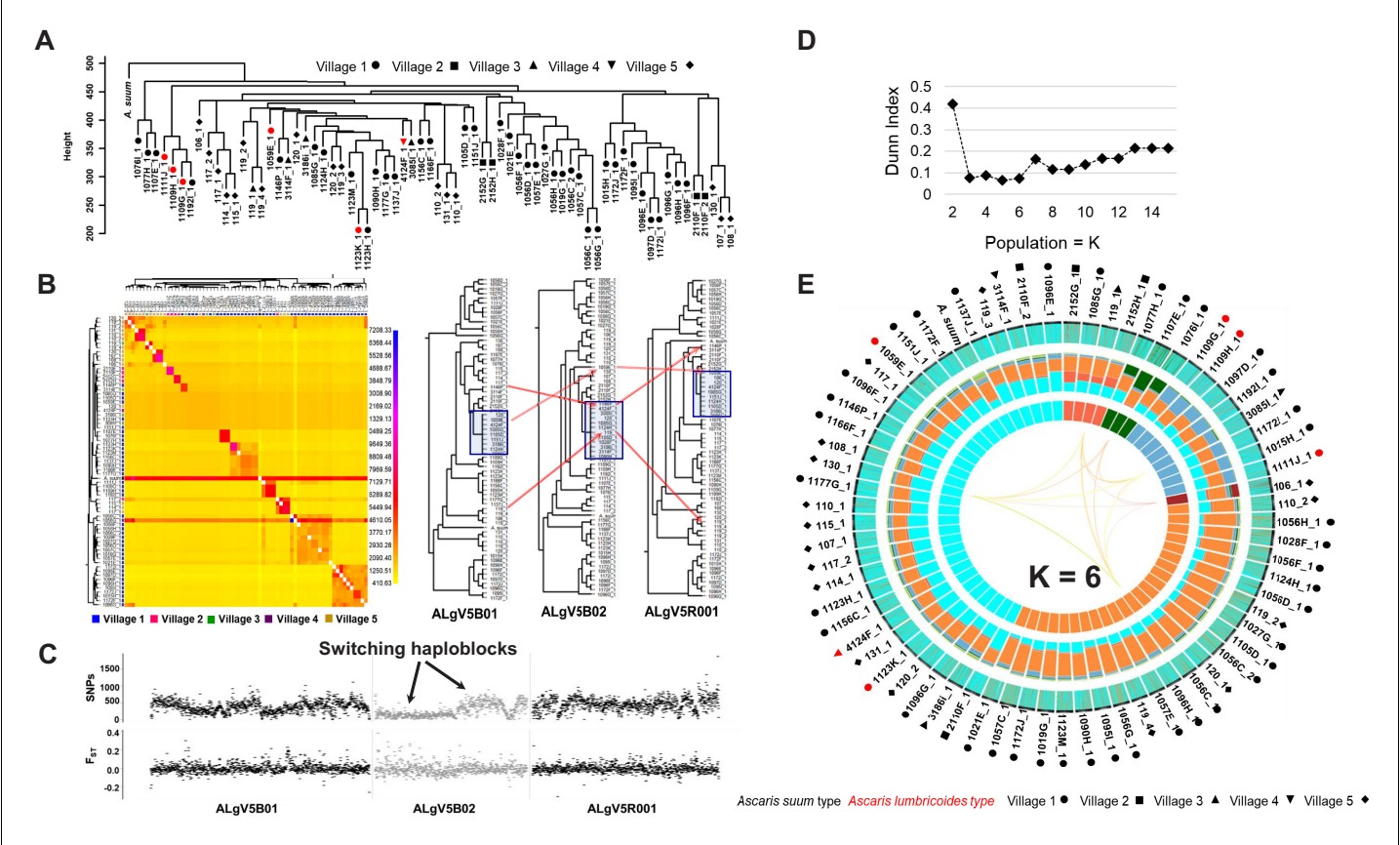

**Figure 4.** Comparative genomics and population genetic structure of *Ascaris*. (**A**) Hierarchy phylogenetic tree of *Ascaris* specimens. Phylogenetic tree was constructed with genome wide SNPs (at 10x coverage) from 68 *Ascaris* specimens, including the *A. suum* reference (outgroup). Height = number of SNPs per site. Red symbol = *A. lumbricoides* mitochondrion genome. Black symbol = *A. suum* mitochondrion genome. Samples were collected from five different villages: Circle = village 1, square = village 2, upside triangle = village 3, downside triangle = village 4, diamond = village 5. (**B**) Heatmap clustering the co-inheritance of ancestral blocks by Bayesian method using genome wide shared haplotype segments among the *Ascaris* genomes. scale = posterior coincidence probability. Hierarchical clustering and phylogenetic relationships are based on percent shared haplotype segments in scaffolds ALgV5B01, ALgV5B02, and ALgV5R001. Red arrows show examples of genetic recombination demonstrated by phylogenetic incongruence in the tree topology based on shared ancestry among blue highlighted specimens (n = 13). (**C**) Pairwise SNPs and $F_{ST}$ estimates in scaffolds ALgV5B01, ALgV5B02, and ALgV5R001 indicate a switching of haplotypes (black arrows), and genetic hybridization among the blue highlighted specimens (n = 13) in the phylogenetic tree depicted in *Figure 2B*. X-axis = total SNPs/10 kb in SNPs plot or $F_{ST}$/10 kb in $F_{ST}$ plot. (**D**) Estimation of the number of ancestral populations (K) based on Dunn Index (*Dunn, 1973*). (**E**) Population genetic structure and admixture clustering analysis of the *Ascaris* genomes obtained by POPSICLE using K = 6 different color hues in the innermost concentric circle of the Circos plot. The middle concentric circle shows the relative percentage of each genetic ancestry within each genome (represented by the color hues for K = 6). The outermost concentric circle shows the genome wide local admixture profile of each worm in 10 kb sliding windows. The following geometric shapes represent villages, and the color for each shape identifies the mitochondrion genome each sample possesses: Black = *A. suum*; red = *A. lumbricoides*; Circle = village 1; square = village 2; upside triangle = village 3; downside triangle = village 4; diamond = village 5.

The online version of this article includes the following figure supplement(s) for figure 4:

**Figure supplement 1.** Admixture clustering and current population genetic structure of Ascaris were determined.

*suum* reference genome, which had many more unique SNPs (*Figure 4A*). Notably, the nuclear genomes from the worms that possessed *A. lumbricoides*-like mitochondrial genomes did not clade separately, indicating that the nuclear genomes were incongruent with the mitochondrial genomes, and likely recombinant. A co-ancestry heatmap was generated among the sympatric *Ascaris*, and this analysis divided the genome into discrete segments and clustered samples along the diagonal based on the greatest number of shared ancestral blocks using the nearest neighbor algorithm from fineSTRUCTURE. The *Ascaris* genomes resolved as 13 clusters that possessed high frequency nearest-neighbor, or shared ancestry, relationships. In contrast, the *A. suum* reference genome and specimen 119_3 were anomalous, likely the result of their excess heterozygosity due in part to elevated

numbers of unique SNPs. Notably, nine worm specimens did not coalesce into a cluster with shared ancestry. Closer examination of these specimens indicated that their phased genomes possessed limited allelic diversity and were highly recombinant (*Figure 4B*). This genetic mosaicism was readily resolved by fluctuating intra-scaffold genealogies established using a sliding-window neighbor-joining topology that identified regions with incongruent tree topologies. See for example the trees generated at the scaffolds ALgV5b01, ALgV5b02, and ALgV5r001. Indeed, the pairwise SNP and $F_{ST}$ estimates for these specimens identified segments where SNP density was low, but $F_{ST}$ was elevated with respect to neighboring segments (see block in ALgV5b02) and the most parsimonious explanation for these results is that recombination of a limited number of distinct alleles had occurred in the regions of increased $F_{ST}$ (*Figure 4B and C*).

To estimate the number of supported ancestries (K) that could be resolved in the *Ascaris* genomes sequenced, we calculated the Dunn index, which supported 3–6 ancestral populations (*Figure 4D*). A gradual increase in the Dunn Index after K = 6 was observed for an ancestral population size between 2 and 15 (*Figure 4D* and *Figure 4—figure supplement 1*). We next used POPSICLE to calculate the number of clades present within each 10 kb sliding window. Local clades were represented with a different color and painted across the genome to resolve ancestry. The SNP diversity plots across the 68 specimens identified three major 'parentage blocks' that were resolved as belonging to ALV5 or were genetically distinct with either both haplotypes sharing the alternate parent (homozygous alternate), or were heterozygous between the two parental haplotypes for the majority of the specimens (*Figure 4E*, middle Circos plot. Color hues cyan, orange, aqua).

To visualize such shared ancestry across the different *Ascaris* specimens at chromosome resolution, a color hue representing a local genetic 'type' present was assigned and integrated to construct haplotype blocks across each chromosome for the ancestries present. Chromosome painting based on shared ancestry revealed a striking mosaic of large haplotype blocks of different admixed color hues, consistent with limited genetic recombination between a low number of parentage haplotypes. These admixture patterns were readily visualized by shared color blocks between different specimens across entire scaffolds including AlgV5R019X (*Figure 5A*) and AlgV5R027X (*Figure 5B*). In low complexity regions such as the left portion of contig ALgV5R019X, only three major haplotypes were resolved (*Figure 5A*). Strikingly, within each of the six clades resolved, all worm specimens showed a limited, mosaic fingerprint of introgressed sequence blocks indicating that recombination has shaped the population genetic structure among the *Ascaris* specimens sequenced. Examples of both chromosomal segregation and recombination were seen. For example, specimens 1107E_1 and 2110F_2 shared the same chromosome at ALgV5R019X, but entirely different chromosomes at ALgV5R027X, whereas specimens 107_1, 108_1 and 2110F_2 were identical except at the subtelomeric end of ALgV5R19X. In this region two admixture blocks were resolved; 107_1 and 2110F_2 remained similar to each other but 108_1 now possessed a sequence block that was shared with specimen 119_3. This extensive chimeric pattern in chromosome painting also closely resembled the genome-wide hierarchy tree (*Figure 5A*). The data support a model in which the specimens are genetic recombinants between *A. suum* and *A. lumbricoides* that are predominantly inbreeding.

## Geographic and demographic correlates of genetic similarity

To examine genetic clustering of worms in individual human hosts, host households and villages, and study time-points, we statistically compared genetic variation within groups (such as within a village) versus between groups (such as between villages). We found significant genetic separation between worms in different villages (*Table 2*, *Figure 6*), although worms from Kenya clustered with worms from around the world based on cox-1, rather than predominantly with each other (*Figure 2A*). This suggests genetic diversity is present in the population of *Ascaris* in these Kenyan villages, which is similar to the diversity of populations of *Ascaris* around the world. It also suggests that a high proportion of *Ascaris* transmission may occur within villages in this Kenyan setting. There was no evidence from this analysis that the 13 worms collected three months after albendazole treatment were any different than the worms collected prior to albendazole treatment (*Table 2*).

To expand on our observations that genetically similar worms are found around the world, but that similar worms cluster within a village, based on our nuclear SNPs data, we plotted genetic distances against geographic distances. Surprisingly, we found no significant correlations between

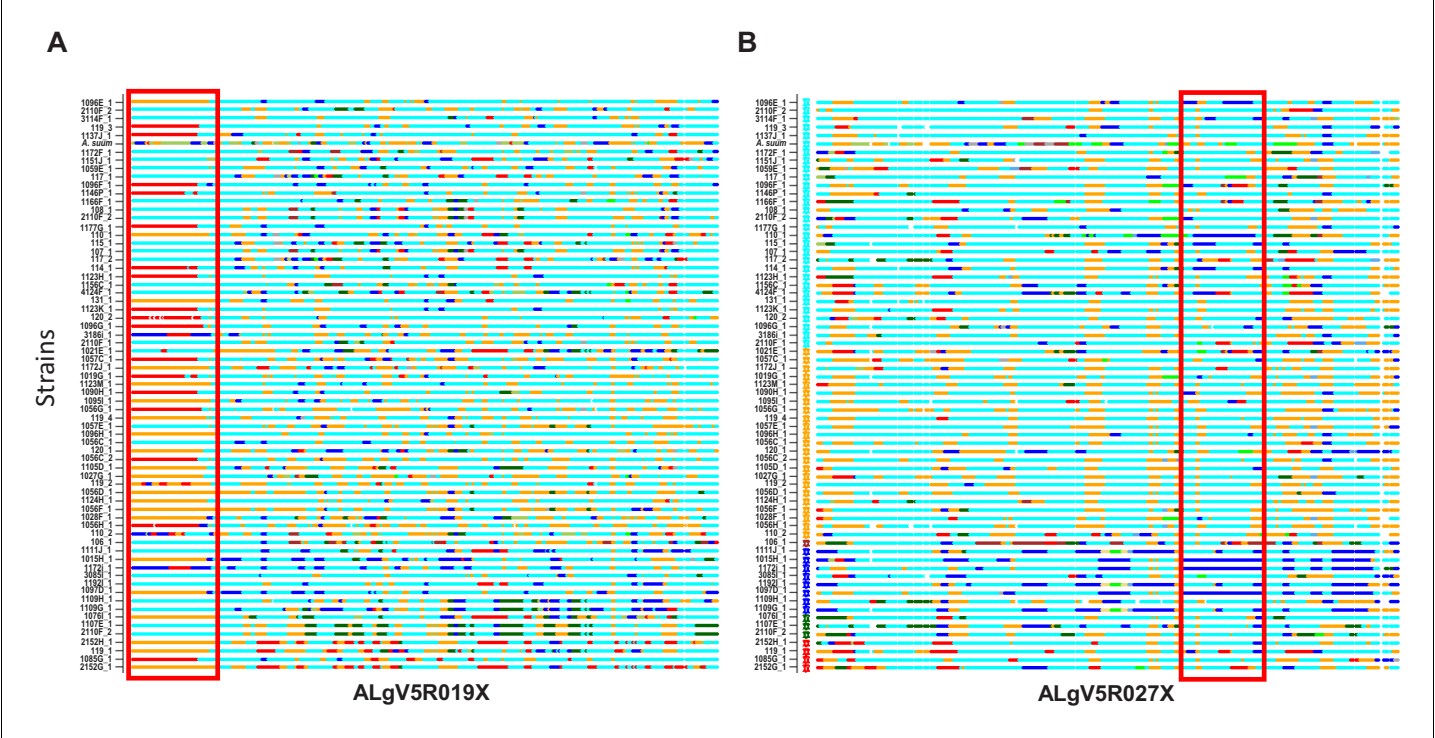

**Figure 5.** Local admixture clustering and genome wide analysis of inheritance of haploblocks of *Ascaris* obtained by POPSICLE (*Shaik et al., 2018*). Based on ancestral population K = 6. X-axis = specimens. Red highlighted box indicates the introgression of large haplotype blocks of defined parentage among the different specimens of *Ascaris* in scaffolds ALgV5R019X (**A**) and ALgV5R027X (**B**). Many examples exist whereby specimens that are in linkage disequilibrium at ALgV5R019X possess different haplotypes in ALgV5R027X (for example 1107E_1 vs. 2110F_2) indicating both segregation as well as recombination in the evolution of the samples. The local admixture patterns reveal extensive genetic hybridization among different strains of *Ascaris*. Color assignment is depicted based on *Figure 4E*.

genetic and geographic distance, neither across all five studied villages nor within the two most heavily parasitized villages (*Figure 6—figure supplement 1*).

## Discussion

In this study, we generated a high-quality reference genome from a single worm presumed to be human *A. lumbricoides*. Our comparative phylogenomic analyses of this new *Ascaris spp.* genome against existing draft genomes of *A. lumbricoides* and *A. suum* suggest that *A. suum* and *A. lumbricoides* form a genetic complex that is capable of interbreeding, which has apparently undergone a recent worldwide, multi-species *Ascaris* population expansion.

Our phylogenetic analysis on the complete mitochondrial genomes (from 68 worms collected from human hosts in Kenya and other available sequences) suggests that the worms collected in Kenya mirror the separation into clade A (worms from pigs in non-endemic regions and humans in endemic regions) and clade B (worms from humans and pigs from endemic and non-endemic regions) described elsewhere (*Cavallero et al., 2013*). It is likely that worms in both these clades are being transmitted from human to human, as pig husbandry is rare in this area of Kenya. Patterns may differ by locality, and it is possible that some of the pig-associated (*A. suum*-like) worms circulating in this human population in Kenya were acquired, perhaps generations ago, by humans who lived in closer proximity to pigs. It is also possible that these worms were acquired from non-human primates (*Nejsum et al., 2010*), or some other *Ascaris* host, rather than from pigs.

However, the SNPs across the whole nuclear *Ascaris* genome provide significantly greater power in understanding *Ascaris* speciation. Importantly, our nuclear genome SNP analysis suggests that the 68 Kenyan *Ascaris* are distributed across multiple clades in a phylogeny based on the nuclear genomes. Overall, data from our study and other studies are consistent with a pattern where hybrid

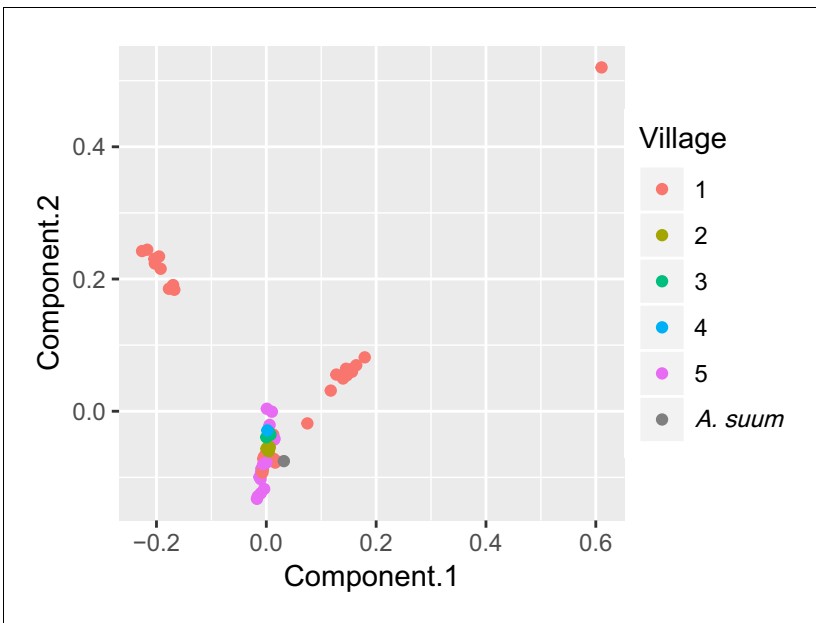

**Figure 6.** PCA plot of worms sequenced for five Kenyan villages. Each point is color-coded by village-of-origin and plotted according to the first and second principal components, based on genome sequences. Worms from village #1 are found in each of three clusters, and two clusters contain only worms from village #1.

The online version of this article includes the following figure supplement(s) for figure 6:

**Figure supplement 1.** Plot of phylogenetic distances compared to geographic distances.
**Figure supplement 2.** Map of Bungoma and West Sang'alo Sub-District.

genotypes in *Ascaris* populations were observed (**Betson et al., 2014**; **Cavallero et al., 2013**; **Criscione et al., 2007**; **Jesudoss Chelladurai et al., 2017**). Our study represents one of the most detailed accounts of mito-nuclear discordance in nematodes echoing patterns seen in another human nematode: *Onchocerca volvulus* (**Choi et al., 2017**). The data in our current study show the occurrence of distinct mitochondrial lineages that could be evidence of early stages of species differentiation. The admixture seen within the nuclear genome, however, appears to disrupt the establishment of defined molecular speciation barriers between the different *Ascaris* lineages. Such patterns have been recorded in other parasites, *including O. volvulus* (**Choi et al., 2017**), the blood fluke *Schistosoma* (**Lawton et al., 2017**) and the protist *Leishmania* (**Kato et al., 2019**). Each of these studies has implicated definitive hosts in the movement of parasites between otherwise isolated populations, allowing interbreeding to take place. It is most likely the historical movement of humans and their domesticated livestock that has mediated the transport of *Ascaris* between localities, allowing for extensive interbreeding as shown by the nuclear genomes and resulting in the discordance observed between the mitochondrial and nuclear genomes in our study.

At a more local scale, the insights into the human transmission dynamics of *Ascaris* showing clustering both within an individual and in villages suggest that villages are appropriate units for interventions and that people are infected with multiple eggs from a single source. These findings are in line with clustering at the village level found in Guatemala (**Anderson et al., 1995**) and at the sub-village level in Nepal (**Criscione et al., 2010**), but not in line with the lack of small-scale geographical structuring found in Denmark, Zanzibar and Uganda (**Betson et al., 2011**; **Betson et al., 2012**; **Nejsum et al., 2005a**). Differences could be a result of different patterns in human and livestock movement (**Betson et al., 2013**).

Although the current genome is, by far, the most continuous assembly for *Ascaris*, it is not a full chromosome assembly due largely to repetitive sequences, in particular 120 bp tandem repeat clusters and long stretches of subtelomeric repeats. Thus, it is possible that mis-assembly in some scaffolds has increased the frequency of mosaicism detected. It is for this reason that the comparative analyses on the nuclear genome was restricted to the largest 50 scaffolds, most of which are at chromosomal resolution, with only minor localized variation due to the repeat clusters. In these high

confidence scaffolds, large haplotype blocks possessing either *A. suum*, *A. lumbricoides* or both parental haplotypes (heterozygous) were readily resolved indicating that the genetic mosaicism observed could not be solely attributed to genome mis-assembly. Ultimately, future studies using ultralong PacBio (*Rhoads and Au, 2015*) or Nanopore (*Branton et al., 2009*) sequencing combined with chromosome conformation capture (Hi-C) techniques (*Belaghzal et al., 2017*) will improve the genome to full chromosome assembly to more accurately resolve the true extent to which recombination has impacted the population genetic structure of the *Ascaris* species genetic complex.

The finding that *A. suum* and *A. lumbricoides* form a genetic complex has important public health implications. Reduced treatment efficacy is not currently a common issue in *Ascaris* infections among humans or pigs (*Levecke et al., 2018*; *Vercruysse et al., 2011*; *Zuccherato et al., 2018*), although low efficacy of benzimidazoles is an issue for *Trichuris trichiura* in humans (*Diawara et al., 2009*; *Furtado et al., 2016*; *Olsen et al., 2009*) and various intestinal nematodes of veterinary importance (*Jaeger and Carvalho-Costa, 2017*; *Kaplan and Vidyashankar, 2012*; *Wolstenholme et al., 2004*). Extensive albendazole use in either human or pig populations could lead to resistance in both populations, if cross-species infections are common and produce fertile offspring. This study suggests that research and public health interventions targeting *A. lumbricoides* and *A. suum* should be more closely integrated, and that extensive work done by the veterinary research community may be highly relevant to mass deworming campaigns that seek to improve human health.

The similarity between *Ascaris* from different countries and from different vertebrate hosts suggests that *Ascaris* infection has spread rapidly around the world, leaving little time for it to differentiate. Taken together, these finding have very important implications for parasite control and elimination efforts that only focus on mass deworming of humans for *Ascaris*. The ability of pig-associated worms to become endemic in human populations indicates that a one-health approach may be necessary for the control of *Ascaris*. The COVID-19 pandemic has highlighted the importance of one health approaches to zoonotic diseases (*Global Burden of Disease, 2020*); we must use a one health approach to ensure that pigs do not serve as a reservoir and potential breeding ground for drug resistance in a parasite that can sustain community transmission in humans (*Webster et al., 2016*).

## Materials and methods

### Worm collection

Worms were expelled as part of a larger study in rural western Kenya described previously (*Easton et al., 2016*, *Easton et al., 2017*). Worms collected from study participants in five villages (*Figure 6—figure supplement 2*) following treatment with 400 mg albendazole were isolated, washed, labeled and stored frozen (−15°C). The villages were near the town of Bungoma, located at N 0.57, E 34.56. Temperatures ranged from 15°C to 30°C and rainfall is 1500 mm on average. Chicken, sheep and cattle farming are common, as is subsistence agriculture and growth of sugar cane as a cash crop. The primary spoken language is Bukusu, a dialect of Luhya.

All samples were stored in Kisumu, from which they were subsequently transported to the KEMRI-CDC offices until they were shipped to the NIH (Bethesda, MD, USA) on dry ice.

### DNA extraction and sequencing

A modified DNA extraction method was developed based on Phenol/Chloroform and Qiagen methods (available on request) and used on 75 samples (*Supplementary file 3*). For the five germline samples, DNA was extracted from the uterus, oviduct or ovary of the worms. For the remaining samples, DNA was extracted from somatic tissue: the body wall or the intestine. Our previous work did not reveal any differences between a variety of somatic samples including the intestine and muscle (*Wang et al., 2012*), thus we do not expect any significant variations in the muscle and intestine genomic DNA used in this study.

Paired-End Genome Libraries – Sixty-eight *A. lumbricoides* DNA samples were sequenced using Illumina HiSeq 2500 (www.illumina.com) short-read paired-end sequencing. DNA was quantified by UV Spec and Picogreen. A 100 ng of DNA based on picogreen quantification was used as template for NGS library preparation using the TruSeq Nano DNA Sample library prep kit without

modification. Primer-dimers in the libraries were removed by additional AMPure beads purification. Sequencing was performed to obtain a minimum genomic depth of 20X coverage for each sample.

Mate-Pair Genome Libraries – Two samples were selected for mate-pair sequencing, based on the quality of the DNA preparation. Three independent DNA isolations (corresponding to what region of the worm or what is the sample for DNA isolation) from specimen '119_2.3' were combined to obtain one µg DNA input. The mate-pair libraries were generated using the Nextera Mate Pair Library Prep Kit, following the gel-free method with the only modification that M-270 Streptavidin binding beads were used instead of M-280 beads. The libraries were amplified for 15 cycles given the low DNA input going into the circularization phase. The mate-pair fragment size averaged 6 kb with a range of 2–10 kb fragments.

## Assembly and annotation of *A. lumbricoides* reference genome

The *A. lumbricoides* germline genome assembly was constructed using the *A. suum* genome as a reference. Briefly, sequencing reads from a single *A. lumbricoides* worm (libraries #8457, #8458, and #8778) were mapped to the *A. suum* germline genome assembly (*Wang et al., 2017*) using BWA (*Li and Durbin, 2009*) to generate BAM and MPILEUP alignment files. The MPILEUP files were processed with a PERL script that replaced all variation sites in the reference genome with the highest allele frequencies in the *A. lumbricoides* sample. *A. suum* genomic regions that represent <5X of *A. lumbricoides* reads coverage were excluded from the assembly. We further polished the genome with additional Illumina sequencing reads using Pilon and its default parameters (*Walker et al., 2014*). The *A. lumbricoides* genome was annotated using the gene models built for *A. suum*, using the annotation transfer tool RATT (*Otto et al., 2011*). The protein coding regions were defined using TransDecoder (https://github.com/TransDecoder/TransDecoder/wiki; *Haas and Papanicolaou, 2016*). To evaluate the gene expression across all stages, we utilized previous RNAseq data from the developmental stages (*Wang et al., 2012*; *Wang et al., 2017*), re-mapped the SRA from adult males, females, L3 and L4 stages (*Jex et al., 2011*) to the current gene models, and quantified the expression using tophat and cufflinks. The re-mapped reads, analyzed by JMP Genomics (SAS) across all the stages and based on the principal component analyses (*Figure 1B*), were grouped as adult male, adult female, L1, L2, L3 (egg L3, liver L3 and lung L3), L4, carcass, muscle, intestine, embryonic (zygote1, zygote2, zygote3, zygote4, 24 hr, 46 hr, 64 hr, 96 hr, 5d, 7d), ovaries (female mitotic region, female early pachytene, female late pachytene, female diplotene and oocyte) and testis (male mitotic region, spermatogenesis, post meiotic region, seminal vesicles and spermatids). Proteome and comparative genomics analyses were done using an in-house pipeline (*Karim et al., 2011*). Automated annotation of proteins was done as described earlier (*Cotton et al., 2017*) and based on a vocabulary of nearly 290 words found in matches to various databases, including Swissprot, Gene Ontology, KOG, Pfam, and SMART, Refseq-invertebrates and a subset of the GenBank sequences containing nematode protein sequences, as well as the presence or absence of signal peptides and transmembrane domains. Signal peptide, SecretomeP, transmembrane domains, furin cleavage sites, and mucin-type glycosylation were determined with software from the Center for Biological Sequence Analysis (Technical University of Denmark, Lyngby, Denmark) (*Duckert et al., 2004*; *Julenius et al., 2005*; *Sonnhammer et al., 1998*). Classification of kinases was done by Kinannote (*Goldberg et al., 2013*). Interproscan (*Jones et al., 2014*) analyses were done using the standalone version 5.34. Allergenicity of proteins were predicted by Allerdictor (*Dang and Lawrence, 2014*), FuzzyApp (*Saravanan and Lakshmi, 2014*) and AllerTOP (*Dimitrov et al., 2014*). Genes that had blast scores < 30% of max possible score (self-blast) in other non-Ascaris nematodes with an e-value greater than 1E-05 were considered as 'unique'. The orthologues of predicted proteome of ALV5 across the publicly available nematode genomes (*Ancylostoma caninum* [*International Helminth Genomes Consortium, 2019*], *Ancylostoma ceylanicum* [*International Helminth Genomes Consortium, 2019*; *Schwarz et al., 2015*], *Ancylostoma duodenale* [*International Helminth Genomes Consortium, 2019*], *Ascaris lumbricoides* [*International Helminth Genomes Consortium, 2019*], *Ascaris suum* [*Jex et al., 2011*; *Wang et al., 2012*; *Wang et al., 2017*], *Brugia malayi* [*Ghedin et al., 2007*], *Caenorhabditis elegans* C. elegans Sequencing [*C. elegans Sequencing Consortium, 1998*], *Dirofilaria immitis* [*Godel et al., 2012*], *Loa loa* [*Desjardins et al., 2013*; *Tallon et al., 2014*], *Necator americanus* [*Tang et al., 2014*], *Onchocerca volvulus* [*Cotton et al., 2017*], *Strongyloides ratti* [*Nemetschke et al., 2010*], *Strongyloides stercoralis* [*Hunt et al., 2016*], *Toxocara canis* [*International Helminth Genomes Consortium,*

*2019*; X.-Q. *Zhu et al., 2015*], *Trichinella spiralis* [*Korhonen et al., 2016*; *Mitreva et al., 2011*], *Trichuris trichiura* [*Foth et al., 2014*], *Wuchereria bancrofti International Helminth Genomes Consortium, 2019*; *Small et al., 2016*) were analyzed using OrthoFinder (*Emms and Kelly, 2015*). The estimated phylogenetic tree generated was graphed using FigTree v1.4.

Further manual annotation was done as required. The data were mapped into a hyperlinked Excel spreadsheet as previously described (*Bennuru et al., 2011*), available in *Supplementary file 2*.

## Read mapping and SNP analysis for whole genome sequences

The Illumina paired-end sequence reads of the 68 *Ascaris* whole genomes were trimmed by removing any adapter sequences with CutAdapt v1.12 (*Martin, 2011*), then low-quality sequences were filtered and trimmed using the FASTX Toolkit (http://hannonlab.cshl.edu/fastx_toolkit/). Remaining reads were then ref-mapped to the *A. lumbricoides* genome ALV5 reference genome (described in this paper) using either Bowtie2 v2.2.9 (*Langmead and Salzberg, 2012*), with very sensitive, no-discordant, and no-mixed settings or using the Burrows-Wheeler Aligner (BWA, v0.7.9) (*Li and Durbin, 2009*) mem in default parameters and then converted into a bam file for sorted with SAMtools (*Li, 2011*). Sorted reads were soft-clipped and marked-duplicated using Picard-1.8.4 (http://broadinstitute.github.io/picard; *Broad Institute, 2020*). Single-nucleotide polymorphisms (SNPs) were obtained using SAMtools (*Li, 2011*) and BCFtools (*Narasimhan et al., 2016*) using the mpileup function and –ploidyfile features and taking chromosomal ploidies into account. SNPs were also determined using Genome Analysis Toolkit (GATK) (*McKenna et al., 2010*). SNPs were called by GATK Haplotype Caller with a read coverage $\geq$10 x, a Phredscaled SNP quality of $\geq$30. Mapping statistics were generated in Perl and Awk.

## Ploidy determination

The ploidy of each specimen was calculated using AGELESS software (http://ageless.sourceforge.net/) by dividing the chromosomes into 10 kb sliding windows and averaging the coverage within each window. The windows with zero coverage were not included in any further analyses due to sequencing noise or repeat regions (*Inbar et al., 2019*).

## Genetic diversity

SNPs, pi (*Nei and Li, 1979*), (*Tajima, 1989*), and $F_{ST}$ (*Dunn, 1973*) values were calculated using VCFtools (*Danecek et al., 2011*) in 10 kb sliding windows and plotted using either Circos (*Krzywinski et al., 2009*) or ggbio (http://bioconductor.org/packages/release/bioc/html/ggbio.html) and VariantAnnotation (http://bioconductor.org/packages/release/bioc/html/VariantAnnotation.html) R packages (v. 3.1.0, URL http://www.R-project.org). The proportions of heterozygous and homozygous SNPs were estimated in 10 kb sliding windows using custom Java scripts to generate histogram plots in Circos (*Krzywinski et al., 2009*). Red and blue colors indicate the presence of 90% or more heterozygous and homozygous SNPs respectively whereas yellow color was assigned otherwise.

## Co-ancestry heatmap

The SNP data (VCF file) was first phased accurately to estimate the haplotypes using SHAPEIT (*Delaneau et al., 2013*) after keeping only biallelic SNPs and loci with less than 80% missing data. Co-ancestry heatmaps were generated using the linkage model of ChromoPainter (*Lawson et al., 2012*) and fineSTRUCTURE (http://www.paintmychromosomes.com) based on the genome-wide phased haplotype data. For fineSTRUCTURE (version 0.02) (*Lawson et al., 2012*), both the burn-in and Markov Chain Monte Carlo (MCMC) after the burn-in were run for 1000 iterations with default settings. Inference was performed twice at the same parameter values.

## Population genetic structure

Population genetic structure was constructed using POPSICLE (*Shaik et al., 2018*) by comparing specimens against the reference sequence ALV5 in 10 kb sliding windows with the number of cluster K = 1 to 15 and then use the Dunn index (*Dunn, 1973*) to calculate the optimal number of clusters. After calculating the optimal number of clusters, POPSICLE assigned each block to the existing or new clades depending on population structure of specimens and the ancestral state of each block

followed by painting in Circos plot (*Krzywinski et al., 2009*) with color assignment based on number of clusters.

## Construction of phylogenetic trees

In order to determine the phylogenetic relationship between samples, we selected 19005 base positions where variants were detected in a representative sample vs the reference (ALV5), and where each sample had at least 20x coverage for each locus. Using this list, the base calls for each sample were pooled together to generate a single multi-sequence fasta file.

Next, both maximum likelihood (ML) trees and bootstrap (BS) trees were generated with a final 'best' tree generated from the best scoring ML and BS trees using RAxML v8.2.10 (*Stamatakis, 2014*). The tree was visualized in FigTree v1.4.3 (http://tree.bio.ed.ac.uk/software/figtree/).

## Permutational multivariate analysis of nuclear phylogeny

Similarity within and between worms from different villages, households, people and time-points was analyzed based on the distance matrix of the patristic distances from the phylogenetic tree described above, using permutational multivariate analysis of variance (Adonis Vegan in R). The distance matrix underlying the phylogenetic tree was analyzed in order to measure the significance and contribution of different factors to variance between samples. Each factor (village, household, host and time-point) was analyzed both separately and sequentially. The sequence chosen was ordered based on significance of each factor when tested individually. Since multiple groupings were considered using the same dataset, multiple comparison corrections were applied. Sample sizes and descriptions of each group are shown in *Table 2*. Similar methods were used to analyze the mitochondrial phylogeny along the same groupings.

## Mitochondrial genome assembly

We assembled mitochondrial genomes using a de novo approach from 68 individual *Ascaris* genomes. For each individual, the *Ascaris* mitochondrial reads in the total DNA sequencing were identified by mapping the *Ascaris* reads to the *A. suum* reference mitochondrial genome (GenBank accession: NC_001327). Adaptor sequences were trimmed prior to de novo assembly. To reduce the complexity of the de novo assembly, we randomly sampled 1000x reads from each individual (the use of higher read coverage often resulted in fragmented scaffolds) and assembled these reads using the SPAdes assembler (*Bankevich et al., 2012*) with continuous k-mer extension from K = 21 to the maximum k-mer allowed (average extended k-mer size = 91). The assembled scaffolds were corrected with the built-in tool in SPAdes to reduce potential assembly artifacts. Next, the assembled scaffolds were aligned to the *A. suum* mitochondrial reference genome using BLAST, the order of the scaffolds was adjusted, and they were joined into a single scaffold. Finally, the gaps in the scaffold were filled using GapFiller (*Boetzer and Pirovano, 2012*) using mitochondrial reads from the same individual to generate a complete mitochondrial genome. Using the same method, we also de novo assembled another five *A. suum* or *A. lumbricoides* mitochondrion genomes from previous studies (see *Supplementary file 8*).

## Analysis of mitochondrial genomes

In order to assess overall evolutionary relationships across the complete mitochondrial genomes, we aligned the genomes using Clustal W and phylogenetic trees constructed using RaxML under the conditions of the general time reversible model (GTR) as described above for the whole genome SNP alignment. Subsequent tree files were formatted in FigTree and MEGA v7. The variation in nucleotide diversity across the mitochondrial genome was measured using sliding window analyses, with a window of 300 bp and a step of 50 bp, using DNAsp v6 (*Rozas et al., 2017*). In order to assess the validity of potential species groupings in the ML phylogenetic tree the *Birky, 2013* X4 ratio was applied to the alignment of the complete mitochondrial genomes including both samples from Kenya and published mitochondrial reference genomes from Tanzania, Uganda, China, USA, Denmark, and the UK. The X4 ratio method of species delimitation compares the ratio of mean pairwise differences between two distinct clades (K) and the mean pairwise differences within each of the clades being compared ($\Theta$). It is considered that if K/$\Theta$ >4 this is indicative of the two clades representing two distinct species. Owing to the fact that two clades are being compared there will be

two separate values of Θ, as per recommendations of *Birky, 2013*, the larger Θ value is used to perform the final ratio calculation as this will provide a more conservative result which ultimately will be less likely to provide a false positive result.

Due to the extensive use of mitochondrial genome data in population genetic analyses of *Ascaris*, several analyses were performed to identify the effect of any population level processes that may be affecting the diversity of the parasites within Kenya. Initially, diversity indices were calculated for each of the genes within the mitochondrial genome across the entire Kenyan data set as well as considering the mitochondrial genome as a whole. In order to account for the diversity within the genic regions, we removed non-coding and tRNA sequences for these analyses. To provide a genealogical perspective of population structure of the Kenya *Ascaris* worm specimens, we constructed the most parsimonious haplotype network based on the protein coding sequences using the TCS algorithm as implemented in PopArt (*Leigh and Bryant, 2015*). Further population genetic analyses were also performed to detect the occurrence of selection on the protein coding genes of the mitochondrial genome and if there were any major departures from neutrality. Standard dN/dS ratios were performed to identify the presence of positive selection where both measures equate to 1 = neutral, >1 = positive selection, <1 = purifying selection. Both Tajima's D and Fu's *Fs* were calculated to identify any substantial departure from neutrality which could be indicative of population expansion events (*Supplementary file 6*). All described analyses were performed using DNAsp6 (*Rozas et al., 2017*). As both cox-1 and nad-4 have been used in the past for epidemiological studies, single gene phylogenies were also constructed as described previously for comparison against the whole mitochondrial genome phylogeny (*Figure 2—figure supplement 2*).

Owing to the extensive use of the cox-1 gene for epidemiological studies the gene was extracted from the complete mitochondrial genomes of Kenya and compared to all other available *Ascaris lumbricoides* and *Ascaris suum* cox-1 sequences housed by NCBI representing populations from across the globe. Haplotype network analyses was performed to produce the parsimonious network using TCS as implemented through PopArt (*Leigh and Bryant, 2015*). This provided a genealogical perspective of population structure and allowed genetic connectivity between the Kenyan samples and samples from other locations to be assessed.

## Acknowledgements

We thank the school children, schoolteachers, and Bungoma administrators for their support. We extend special thanks to all the members of the study team: Bungoma County Hospital, Siangwe, Siaka, Sang'alo, Nasimbo and Ranje village administrators and Community Health Workers. Particular thanks to Dr. Charles S Mwandawiro, Prof. Sammy Njenga, and Dr. Jimmy H Kihara (KEMRI), and Dr Simon J Brooker (BMGF) for making the fieldwork possible in Kenya, and for their invaluable scientific and logistical advice.

## Additional information

### Competing interests
Roy Anderson: RMA was a Non-Executive Director of GlaxoSmithKline (GSK) during the period of worm collection in Kenya. GSK played no role in the funding of this research or this publication. The other authors declare that no competing interests exist.

### Funding

| Funder | Grant reference number | Author |
| --- | --- | --- |
| National Institutes of Health | AI114054 | Richard E Davis |
| National Institutes of Health | AI125869 | Jianbin Wang |
| Bill and Melinda Gates Foundation | | Roy Anderson |
| National Institutes of Health | | Thomas B Nutman |
| Wellcome Trust | KEMRI | Roy Anderson |

| London Centre for Neglected Tropical Disease Research | Roy Anderson |

The funders had no role in study design, data collection and interpretation, or the decision to submit the work for publication.

## Author contributions

Alice Easton, Conceptualization, Data curation, Formal analysis, Investigation, Visualization, Methodology, Writing - original draft, Project administration, Writing - review and editing; Shenghan Gao, Formal analysis, Investigation, Methodology; Scott P Lawton, Formal analysis, Visualization, Methodology, Writing - original draft, Writing - review and editing; Sasisekhar Bennuru, Asis Khan, Formal analysis, Investigation, Visualization, Methodology, Writing - original draft; Eric Dahlstrom, Data curation, Formal analysis, Investigation, Visualization, Methodology, Writing - original draft, Writing - review and editing; Rita G Oliveira, Stella Kepha, Conceptualization, Data curation; Stephen F Porcella, Data curation, Investigation; Joanne Webster, Conceptualization, Supervision, Writing - review and editing; Roy Anderson, Conceptualization, Resources, Supervision, Funding acquisition, Methodology; Michael E Grigg, Formal analysis, Supervision, Investigation, Visualization, Methodology, Writing - original draft, Writing - review and editing; Richard E Davis, Conceptualization, Supervision, Funding acquisition, Investigation, Methodology, Writing - original draft, Writing - review and editing; Jianbin Wang, Conceptualization, Funding acquisition, Formal analysis, Supervision, Investigation, Visualization, Methodology, Writing - original draft, Writing - review and editing; Thomas B Nutman, Conceptualization, Resources, Supervision, Funding acquisition, Investigation, Visualization, Methodology, Writing - original draft, Project administration, Writing - review and editing

## Author ORCIDs

Alice Easton (iD) https://orcid.org/0000-0002-4476-9415
Scott P Lawton (iD) https://orcid.org/0000-0003-4055-6524
Sasisekhar Bennuru (iD) https://orcid.org/0000-0002-6117-742X
Eric Dahlstrom (iD) https://orcid.org/0000-0002-9333-3060
Richard E Davis (iD) https://orcid.org/0000-0003-2242-3234
Jianbin Wang (iD) https://orcid.org/0000-0003-3155-894X
Thomas B Nutman (iD) https://orcid.org/0000-0001-6887-4941

## Ethics

Human subjects: This study was approved by the Ethics Review Committee of the Kenya Medical Research Institute (Scientific Steering Committee protocol number 2688) and the Imperial College Research Ethics Committee (ICREC_ 13_1_15). Informed written consent was obtained from all adults and parents or guardians of each child. Minor assent was obtained from all children aged 12-17. Anyone found to be infected with any STH was treated with 400 mg ALB during each phase of the study, and all previously-untreated village residents were offered ALB at the end of each study phase.

## Decision letter and Author response

Decision letter https://doi.org/10.7554/eLife.61562.sa1
Author response https://doi.org/10.7554/eLife.61562.sa2

# Additional files

## Supplementary files

• Supplementary file 1. Characteristics of genome assemblies. Reference *A. lumbricoides* genomes generated as part of this study (1 and 3) are compared with reference genomes for *A. suum* generated previously (2 and 4).

• Supplementary file 2. Proteome annotation. While ~94.6% of the genes can be transferred to both genomes, over 20% of the transferred genes are only partial matches and are fragmented supporting the view that the de novo and semi de novo *A. lumbricoides* assemblies are highly fragmented.

• Supplementary file 3. Description of worm from which each sample was sequenced. The sex of the worm (based on morphological identification) and the part of the worm (germline vs somatic) is listed. Some hosts donated multiple worms.

• Supplementary file 4. cox-1 haplotype list.

• Supplementary file 5. X4 ratio analyses of Clades A and B using complete mitochondrial genomes used to construct the phylogeny in *Figure 2b*.

• Supplementary file 6. Demographic analyses using Tajima's D and Fu's F statistic across complete mitochondrial genomes as a detection for the signature of population expansion events. Whether all sequences collected globally, or just sequences collected in Kenya as part of this study were examine, the Tajima's D value was negative and significant (indicating an excess of low frequency polymorphisms) and the Fu's Fs was positive but not significant (potentially indicating a deficiency in diversity as would be expected in populations that has recently undergone a bottle neck event).

• Supplementary file 7. Number of heterozygous and homozygous SNPs in each of the 68 worms from Kenya sequenced.

• Supplementary file 8. Reference mitochondrion genomes.

• Supplementary file 9. Supplement to *Table 2* using alternative measures of phylogenetic distance.

• Transparent reporting form

## Data availability

Data are available under the National Center for Biological Information (NCBI) BioProject numbers; PRJNA511012 for raw sequencing data, and PRJNA515325 for the genomic assembly. Links to all genome assemblies are available at: All (https://s3.amazonaws.com/proj-bip-prod-publicread/his-omics/ALv5/Genome_Assembly/Genome_assemblies.tar.gz), De Novo (https://s3.amazonaws.com/proj-bip-prod-publicread/his-omics/ALv5/Genome_Assembly/AL-version0-genome-assembly.fasta.gz), Semi-De Novo (V1) (https://s3.amazonaws.com/proj-bip-prod-publicread/his-omics/ALv5/Genome_Assembly/AL-version1-genome-assembly.fasta.gz), V2 – (https://s3.amazonaws.com/proj-bip-prod-publicread/his-omics/ALv5/Genome_Assembly/AL-version2-genome-assembly.fasta.gz), V3 – (https://s3.amazonaws.com/proj-bip-prod-publicread/his-omics/ALv5/Genome_Assembly/AL-version3-genome-assembly.fasta.gz), V4 – (https://s3.amazonaws.com/proj-bip-prod-publicread/his-omics/ALv5/Genome_Assembly/AL-version4-genome-assembly.fasta.gz), V5 – (https://s3.amazonaws.com/proj-bip-prod-publicread/his-omics/ALv5/Genome_Assembly/AL-version5-genome-assembly.fasta.gz), Mitochondrial – (https://s3.amazonaws.com/proj-bip-prod-publicread/his-omics/ALv5/Genome_Assembly/mitochondrial_genomes.tar.gz).

The following datasets were generated:

| Author(s) | Year | Dataset title | Dataset URL | Database and Identifier |
|---|---|---|---|---|
| Easton A, Gao S, Dahlstrom E, Porcella SF, Davis RE, Wang J, Nutman TB | 2018 | 68 Ascaris lumbricoides WGS reads from Bungoma County, Kenya | https://www.ncbi.nlm.nih.gov/bioproject/PRJNA511012 | NCBI BioProject, PRJNA511012 |
| Easton A, Gao S, Dahlstrom E, Porcella SF, Davis RE, Wang J, Nutman TB | 2019 | Reference quality genome assembly of Ascaris lumbricoides from the Bungoma region of Kenya | https://www.ncbi.nlm.nih.gov/bioproject/PRJNA515325 | NCBI BioProject, PRJNA515325 |

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
