## [Decision Letter]

**Acceptance summary:**

This manuscript provides an answer to a long held question of whether *Ascaris* worms predominately infecting pigs or humans are distinct species. The manuscript provides a substantial new body of nucleotides sequences from >60 individual *Ascaris* worms collected from ~60 infected people in five villages of rural Kenya. These sequences were compared with available sequences from around the globe for the human large roundworm, *Ascaris lumbricoides* and the pig parasite *Ascaris suum*. The reported data indicate that the new genomes represent a worm that is a chimeric of the pig and human *Ascaris* species, with further evidence provided that such chimeric worms were inbreeding within the human participants. These findings provide new evidence that the worms infecting humans should be considered as an *Ascaris* species continuum

**Decision letter after peer review:**

Thank you for submitting your article "Extensive hybridization between pig and human *Ascaris* identifies a highly interbred species complex infecting humans" for consideration by *eLife*. Your article has been reviewed by two peer reviewers, and the evaluation has been overseen by a Reviewing Editor and Dominique Soldati-Favre as the Senior Editor. The following individual involved in review of your submission has agreed to reveal their identity: Paul J Brindley (Reviewer #1).

The reviewers have discussed the reviews with one another and the Reviewing Editor has drafted this decision to help you prepare a revised submission.

Summary:

This manuscript addresses a question that has debated for decades and is both timely and interesting. It deals with human ascariasis and confirms a long held hypothesis that the pig *Ascaris*/human *Ascaris* phenomenon constitutes an *Ascaris* species continuum.

It provides a substantial new body of nucleotides sequences from the nuclear – both somatic and germline genome and the mitochondrial genome of >60 individual *Ascaris* worms collected from ~60 infected people in five villages of rural Kenya. The investigators present findings from genomic, proteomic and mt comparisons and phylogenetic trees, and compared the new sequences with available sequences from around the globe for the human large roundworm, *Ascarislumbricoides* and the pig parasite *Ascarissuum*. They now report that most of these Kenyan worms exhibited mt genomes more similar to those described elsewhere to *A. suum* than to *A. lumbricoides*, and that the nuclear genome included mixed *A. suum*-like and *A. lumbricoides* like regions, such that the genomes could be described as chimeric of the pig and human *Ascaris* species. The investigators consider that their data indicate that these worms of the chimeric genotypes were predominantly inbreeding within each of the study villages.

The paper provides a novel characterization of the *Ascaris* parasites circulating in the study region. The phylogenetic trees from mt, from nuclear genome, and from protein encoding genes, and other details, together provided cogent evidence of the chimeric (*lumbricoides-suum*) nature of the *Ascaris* parasitizing the participants of the study. This study looks to have been carried out in a professional manner without obvious scientific or technical flaws. The voluminous amount of novel new nucleotide sequences represent a marked advance in genetic information for this major neglected human pathogen.

Essential revisions:

1) Provide an adequate description of the study site. The readers are directed for this to Easton et al., 2016, Easton et al., 2017. Yet, descriptions there are themselves cursory, and refer to a report from a decade earlier (Pullan et al., 2011). Provide – even in supporting materials – an informative, illustrative description of the study site (consider providing the map to which you indicate is present in Easton et al., 2016 even though the Easton et al., 2016 paper does not appear to include a map…, and also geographical coordinates of these five villages, epidemiological and demographical characteristics, livestock farming (if relevant) characteristics, and so forth.

2) In the Discussion, state clearly why you consider that these new findings – *Ascarislumbricoides* and *A. suum* jointly represent a species complex – is novel, given the earlier papers that have stated similar conclusions, e.g. Leles et al., 2012; Monteiro et al., 2019; Sadaow et al., 2018.

3) With respect specifically to the 'Gene content and *Ascaris* proteome' section of the Results, consider revising to more clearly indicate what the findings revealed and what they mean: What did the predicted proteome details indicate about the chimeric (*lumbricoides-suum*) nature of the *Ascaris* parasitizing the participants of the study? Or were the findings employed only for the phylogenetic tree presented as Figure 1C.

4) Given your use (several times) of the term anthroponotic, which indicates that Homo sapiens is the reservoir of the '*Ascarislumbricoides*-*suum*' chimera – species complex, comment on the host range of this *Ascaris* species beyond humans and pigs. Provide supporting citations.

5) Elaborate of what you recommend for the control and treatment of parasitism by *Ascaris* based on your brief comments advising the need for a One Health approach. Provide supporting references to buttress the recommendation.

6) Please revise the title, a reader might read the title and understand that experimental hybridization might have been undertaken. A suggested modification: "Molecular evidence of.…. indicates…."

7) The narrative of the manuscript is very nice. However, there are some issues that could be improved. First, the text contains typographical and grammatical errors that should be eliminated – also the errors in the reference list and the citations need to be corrected. Please meticulously edit the manuscript. Second, "isolates" and "strains" are confusing terms, if one is referring to individual worms (which maybe the case). As it is written, DNA samples were isolated from muscle or intestine from single worms, thus using the terms "worm" or individual" or "individual worm or specimen" is recommended

8) Genomic DNA was not consistently isolated from muscle, but rather from muscle or intestine; might this have an impact on the nuclear genomic sequences of individuals? In other words, can one be certain that there is no genomic variation within individuals (excluding germline)?

---

## [Author Response]

Essential revisions:1) Provide an adequate description of the study site. The readers are directed for this to Easton et al., 2016, Easton et al., 2017. Yet, descriptions there are themselves cursory, and refer to a report from a decade earlier (Pullan et al., 2011). Provide – even in supporting materials – an informative, illustrative description of the study site (consider providing the map to which you indicate is present in Easton et al., 2016 even though the Easton et al., 2016 paper does not appear to include a map…, and also geographical coordinates of these five villages, epidemiological and demographical characteristics, livestock farming (if relevant) characteristics, and so forth.

We have revised the manuscript to include a more detailed description of the study site, and a new supplementary figure with a map. The new description that appears in the Materials and methods is as follows:

“Worm collection (and study site)

Worms were expelled as part of a larger study in rural western Kenya described previously (Easton et al., 2016, 2017). Worms collected from study participants in five villages (Figure 6—figure supplement 2) following treatment with 400 mg albendazole were isolated, washed, labelled and stored frozen (-15 C). […] All samples were stored in Kisumu from which they were subsequently transported to the KEMRI-CDC offices until they were shipped to the NIH (Bethesda, MD, USA) on dry ice.”

2) In the Discussion, state clearly why you consider that these new findings – Ascaris lumbricoides and A. suum jointly represent a species complex – is novel, given the earlier papers that have stated similar conclusions, e.g. Leles et al., 2012; Monteiro et al., 2019; Sadaow et al., 2018.

While the conclusions have been previously suggested, the current manuscript and data provide comprehensive and more definitive support for a species continuum than previous work. We tried to put our findings in the context of these and similar works in the Introduction (third paragraph) and Discussion. Leles is already quoted in the Introduction. We feel that the papers we quoted in the Discussion (Cavallero et al., 2013 and Nejsum et al., 2010) were field-defining. We appreciate the suggestion of the papers by Monteiro and Sadaow, but elected to include them in the broader list of manuscripts quoted in the Introduction, rather than adding them into the Discussion. However, we thank the reviewers for bringing these two papers to our attention, especially because they had access to excellent samples from both human and pig samples (thus making them different from our own setting as well, where there were no pigs from which to collect samples). Thus we have added the references in the verbiage that is appended below.

“Some studies have shown that cross-species transmission occurs between pigs and humans living in close proximity (Betson et al., 2014; Nejsum et al., 2005; Miller et al., 2015; Anderson, 1995; Zhu et al., 1999; Peng and Criscione, 2012; Takata, 1951; Sadaow et al., 2018; Monteiro et al., 2019)”.

3) With respect specifically to the 'Gene content and Ascaris proteome' section of the Results, consider revising to more clearly indicate what the findings revealed and what they mean: What did the predicted proteome details indicate about the chimeric (lumbricoides-suum) nature of the Ascaris parasitizing the participants of the study? Or were the findings employed only for the phylogenetic tree presented as Figure 1C.

To make our goals more clear (and to highlight these goals ) we have added the following sentence to the Results:

“Our aims were to highlight the phylogenetic relationship with other helminths and between *Ascaris* spp., to provide potential targets for future diagnostics to differentiate between nematodes and even between pig and human *Ascaris*, and to detail the potential functions of hypothetical or unknown proteins in the *Ascaris* genome.”

4) Given your use (several times) of the term anthroponotic, which indicates that Homo sapiens is the reservoir of the '*Ascaris lumbricoides-suum*' chimera – species complex, comment on the host range of this Ascaris species beyond humans and pigs. Provide supporting citations.

We have added the following text:

“*Ascaris* sp. infections also occur naturally in monkeys and apes, and *Ascaris* sp. eggs are sometimes found in the feces of dogs but this is likely a result of coprophagy by the dogs, rather than due to infection (https://www.cdc.gov/parasites/ascariasis/biology.html).”

5) Elaborate of what you recommend for the control and treatment of parasitism by Ascaris based on your brief comments advising the need for a One Health approach. Provide supporting references to buttress the recommendation.

In short, we believe that the fact that a pig-origin worm can spread human-to-human is direct evidence that controlling *Ascaris* in humans will never be sufficient if animals remain a reservoir. We have added two key references to bolster this point as follows:

“The ability of pig-associated worms to become endemic in human populations indicates that a one-health approach may be necessary for the control of *Ascaris*. The COVID-19 pandemic has highlighted the importance of one health approaches to zoonotic diseases (Emerging zoonoses: A one health challenge, 2020); we must use a one health approach to ensure that pigs do not serve as a reservoir and potential breeding ground for drug resistance in a parasite that can sustain community transmission in humans (Webster et al., 2016)”.

6) Please revise the title, a reader might read the title and understand that experimental hybridization might have been undertaken. A suggested modification: "Molecular evidence of.…. indicates…."

The title has been revised to:

“Molecular evidence of extensive hybridization between pig and human *Ascaris* indicates a highly interbred species complex infecting humans”

7) The narrative of the manuscript is very nice. However, there are some issues that could be improved. First, the text contains typographical and grammatical errors that should be eliminated – also the errors in the reference list and the citations need to be corrected. Please meticulously edit the manuscript.

We have hand checked this final version and done our best to be consistent throughout and following the appropriate formatting for eLife. We have examined the format of all the references individually and tried to ensure that they are all formatted correctly.

Second, "isolates" and "strains" are confusing terms, if one is referring to individual worms (which maybe the case). As it is written, DNA samples were isolated from muscle or intestine from single worms, thus using the terms "worm" or individual" or "individual worm or specimen" is recommended

Where “isolate” and “strain” occur, we have changed this to worm/specimen/sample and in some cases deleted the word as extraneous. One instance of strain remains, where other words would be incorrect.

8) Genomic DNA was not consistently isolated from muscle, but rather from muscle or intestine; might this have an impact on the nuclear genomic sequences of individuals? In other words, can one be certain that there is no genomic variation within individuals (excluding germline)?

We were not able to compare muscle and intestinal DNA as part of this study. However, we believe that the genomic vs. somatic distinction was the most meaningful. We have added the following:

“Our previous work did not reveal any differences between a variety of somatic samples including the intestine and muscle (Wang, et al., 2012), thus we do not expect any significant variations in the muscle and intestine genomic DNA used in this study.”